# Psychological Risk Factors for the Development of Restrictive and Bulimic Eating Behaviors: A Polish and Vietnamese Comparison

**DOI:** 10.3390/nu13030910

**Published:** 2021-03-11

**Authors:** Bernadetta Izydorczyk, Ha Truong Thi Khanh, Małgorzata Lipowska, Katarzyna Sitnik-Warchulska, Sebastian Lizińczyk

**Affiliations:** 1Institute of Applied Psychology, Faculty of Management and Social Communication, Jagiellonian University, 30-374 Krakow, Poland; bernadetta.izydorczyk@uj.edu.pl (B.I.); katarzyna.sitnik-warchulska@uj.edu.pl (K.S.-W.); octans@wp.pl (S.L.); 2Faculty of Psychology, University of Social Sciences and Humanities, Vietnam National University, Hanoi 100000, Vietnam; ttkha@vnu.edu.vn; 3Institute of Psychology, University of Gdansk, 80-309 Gdansk, Poland

**Keywords:** eating disorders, body image, psychological factors for eating disorders, cross-cultural psychology, Westernization

## Abstract

The aim of this paper is to identify psychological factors which are culture specific or common predictors for restrictive and bulimic behaviors towards eating for young women raised in different cultures. The study included 661 young women from Poland (*n* = 233) and Vietnam (*n* = 428). Subjects filled-in the Eating Disorders Inventory (EDI-3) and the Multidimensional Body-Self Relations Questionnaire-Appearance Scales (MBSRQ-AS), and body measurements were collected to calculate anthropometric indices. Women form Vietnam were less satisfied with their appearance than were their Polish peers, but Vietnamese showed a lower level of preoccupation with being overweight and fear of obesity. Intercultural differences indicate that Vietnamese women show greater intensities for psychological variables, connected with restrictive and bulimic eating behaviors, verified in the research model: low self-esteem, personal alienation, interpersonal insecurity, interpersonal alienation, emotional dysregulation, interoceptive deficits, perfectionism and asceticism, and anxiety.

## 1. Introduction

### 1.1. Epidemiology of Eating Disorders

Eating disorders are diagnosed all over the world, despite geographical differences, climates, culinary traditions, and even societal wealth levels [1,2]. Epidemiological data confirm the growing significance of restrictive and bulimic eating behaviors in European countries [3,4], including Poland [5,6], North America [7,8], Latin America [9], Arabian countries [10,11], Africa [12], and Asia [13,14,15,16], including Vietnam [17,18,19]. Increasing eating disorder rates are mainly seen in women [20,21,22], although increasing rates are also reported for men [21,23,24,25]. In terms of psychological predictors of abnormal health, restrictive and bulimic (compulsive) eating behaviors are an important factor in the development of eating disorders such as anorexia, bulimia, and compulsive overeating [26,27,28,29,30,31]. Cross-cultural comparison of young women raised in different cultures can significantly deepen the diagnostic potential for psychosocial risk factors in terms of eating disorders specific to these cultures. The growing number of young women are involved anti-health behavior to reduce body weight, such as over-exercising, inducing vomiting or using laxatives indicates a growing need for research on psychosocial risk factors for these behaviors in various populations. The increase in eating disorders and self-destructive behaviors towards the body indicates that the growing risk of these disorders applies not only to women raised in Western countries, but also to citizens of emerging economies and developing countries that have experienced industrialization, modernization, and Westernization, including several Central European and Asian countries [32,33,34,35,36]. Although a research review has shown that eating disorders are increasingly diagnosed in Asia, Europe, and America, but different research approaches have been used, and only some of the reported research results have described potential psychological and socio-cultural predictors for eating disorders. Therefore, it is important to conduct new and different population studies in which multifactorial models of psychological and sociocultural risk factors for anti-health behaviors related to the development of eating disorders in Western and Asian cultures are verified. The research in this article considers Vietnamese and Polish women who grew up in Asian and European emerging countries. Regarding popular culture in America, Europe, and Asia, these areas often promote a common process of unifying socio-cultural standards. Poland’s Westernization took place after the communist period [37] and Vietnam’s Westernization and industrialization began after renovation in the mid-1980s [38,39,40]. According to Stigler et al. [41] Westernization causes very similar changes in different countries. Taking into account the potential differences and similarities resulting from upbringings in various environmental circumstances in terms of in the shaping psychological and social profiles, these potential differences and similarities should be taken into account in studies of women from both Asian and European countries. For evidence-based practice, the measurement of anti-health behaviors towards the body and eating requires constant verification with various European, Asian, and American populations in order to extend the knowledge of universal and culturally specific and psychological predictors of eating disorders.

### 1.2. Psychological Risk Factors for Eating Disorders with Cross-Cultural Ccomparisons

Under the influence of globalization and Westernization, unified national standards of beauty have formed in Asian, Arabian, European, and American countries [42]. Despite significant uniformity in appearance ideals, some cross-cultural differences exist [43,44]. In developing countries, both in Europe and Asia, e.g., Poland and Vietnam, the pursuit of thinness is especially significant [17,18,33,45,46,47,48]. A literature review has confirmed the importance of socio-cultural influences in terms of body image standards promoted by the media on the development of restrictive and/or bulimic eating behaviors [32,49,50,51,52,53,54]. Dissatisfaction with one’s body appearance is particularly common for young women [21,55,56]. Apart from the influence of socio-cultural factors on the ideal body image, the measurement of psychological factors should be taken into account in the process of searching for factors that influence or mediate the development of anti-health behaviors towards eating. For this reason, researchers must test models with variable psychological factors, in addition to sociocultural factors, in order to elucidate a multifaceted psychological model that describes predictors of restrictive and bulimic behavior development [30,50,57]. A review of the studies conducted on populations of women in European and Asian countries has confirmed that body dissatisfaction is one of the important psychological risk factors for developing eating disorders or behaviors consisting of an excessive restrictive pursuit of thinness and the use of bulimic behavior [2,15,17,32]. The results of some studies have shown significant correlations between restrictive and compulsive (bulimic) eating behaviors and psychological factors and personality traits [32,58]. Some studies have indicated that people suffering from eating disorders are characterized by a specific profile of psychological traits, and that these traits are worth paying attention to when performing a psychological diagnosis when treating anorexia or bulimia [59]. Another review has indicated the diverse methods used to assess the frequency of eating disorders and the different indicators of morbidity in these studies, also depending on the measurement tools [1,20]. Among the basic psychological factors of people with specific disorders, nutrition researchers most often mention dissatisfaction with the body (described by appearance evaluation, appearance orientation satisfaction with individual areas of the body, body weight, and the fear of gaining weight), emotional dysregulation, interoceptive deficits, maturity fears, low self-esteem, personal and, interpersonal alienation, interpersonal insecurity, perfectionism, and asceticism [2,32,35,60,61,62,63,64,65,66,67,68,69,70,71,72]. Taking into account the large number of psychological risk factors for eating disorders and the methodological standards for measuring these factors, the authors of this article, while building their own research model, referred to recognized indicators with high statistical validity for measuring eating disorder profiles in terms of psychological features [60].

### 1.3. Research Objective, Variables, and Research Questions

This article focuses on Polish and Vietnamese women. The main goal of this study was to determine which psychological factors for eating disorders are specific predictors (only for Polish and Vietnamese women), including the elucidation of common predictors for restrictive and bulimic behaviors towards eating for the considered women in Vietnam (Asian culture) and Poland (European culture). In order to achieve this goal, first, the intercultural similarities and differences between young Polish and Vietnamese women were elucidated in terms of the relevance of each variable in the research model. These variables included the average intensity of psychological factors describing psychological dispositions and features (explanatory variable), body image (intermediary variable), and restrictive and bulimic behaviors (explained variable). The second goal of the research was to determine, among the psychological factors in the research model, whether they exist and which are universal and specific for Polish and Vietnamese cultures in terms of psychological predictors for the development of restrictive and bulimic behaviors towards eating. No comparative studies exist for psychological predictors for the development of restrictive and bulimic behaviors for Polish and Vietnamese women, and there have been no comparative studies on the features describing the body images of Polish and Vietnamese women. In Polish and Vietnamese research, there also is no definition of the strength of the indirect and direct influences of a variable body image (understood as a multi-element structure defining emotional and cognitive aspects of the attitude towards the body) on the formation of restrictive and bulimic behaviors in Polish and Vietnamese women. After finding no comparative studies, the authors of this article chose to not refer to comparisons between the variables discussed in the study model with those of other authors. Moreover, literature sources indicate that the psychological factors presented in this article, as potential predictors of restrictive and bulimic behaviors, have been analyzed in other studies using various methods and most often are only concerned with the measurement of specific psychological characteristics [49,60,67] or socio-cultural variables [6,50,57]. The explanatory variable in this study was defined as a multi-element structure of psychological factors, which, according to various studies, may potentially explain restrictive and bulimic eating behaviors. The explanatory variable pertains to the effects of the following psychological features and dispositions:Low self-esteem—an indicator of self-esteem level;Personal alienation—describes level of reflectiveness and emotional emptiness;Interpersonal insecurity—allows to determine the intensity of difficulties in expressing personal thoughts and feelings when other people are present, and the tendency to isolate oneself;Interpersonal alienation—refers to the level of disappointment, alienation, and lack of trust in relationships;Emotional dysregulation—an indicator describing the level of intensity of mood instability, impulsiveness, recklessness, anger, and a tendency towards self-destruction;Interoceptive deficits—an indicator describing one’s level of confusion in the accurate recognition of emotional states and stimuli from one’s own body;Perfectionism— an index of the intensity of the need for the highest possible accomplishment and the tendency to possess the maximal achievable standards for personal achievement;Asceticism—an indicator that describes tendency to seek purity through striving for spiritual ideals such as self-denial, self-discipline, self-restraint, and self-control. This concept encompasses the control of needs and drives, as well as it assesses positive connotations associated with reaching purity by the means of restraint, guilt, and shame regarding pleasure;Maturity fears—an indicator describing the strength of a person’s longing for the return to the safety of childhood. It is also associated with the fear of psychosexual puberty.

The second explanatory variable was body image, which was defined as a multi-element structure describing the emotional and cognitive attitude towards the body and appearance as revealed by the appearance evaluation, appearance orientation, body area satisfaction, own body weight, and perceived level of anxiety in terms of weight gain.

An additional independent variable was the BMI (body mass index), measured according to the weight in kilograms divided by the square of the height in meters.

The explained variable was restrictive and bulimic behavior towards eating. Restrictive behaviors were defined responses and behaviors related to eating restriction and excessive pursuits of thinness due to fear and guilt of gaining weight while maintaining a normal BMI. Bulimic behavior was defined as a person’s reactions and behaviors involving excessive overeating and the use of body laxatives (i.e., vomiting) (Figure 1).

The following research questions are considered here:Is there a difference between Polish and Vietnamese women in terms of the psychological factor (dispositions and traits) intensity in the research model and components describing the body image of the surveyed women, and additionally the intensity of restrictive and bulimic behaviors used by the surveyed women?What psychological factors (explanatory variable) are important for the development of restrictive and bulimic eating behaviors in Polish and Vietnamese women?What is the role of the body image in relation to the generation of restrictive and bulimic eating behaviors in young Polish and Vietnamese women? Is it an intermediary variable in their formation? If so, which aspects describing body image in the research model mediate the formation of restrictive and/or bulimic behaviors?Which of the psychological factors verified in the research model have significant direct impacts, and which are not significant for the development of restrictive and bulimic behaviors towards eating for young Polish and Vietnamese women?

## 2. Materials and Methods

### 2.1. Participants

Groups were selected through purposeful sampling. The following inclusion criteria were used: age (19–25 years old), Polish or Vietnamese nationality, residence in Poland or Vietnam for specific groups, no presence of treatment due to any eating disorder, no apparent disability or distortion of physical appearance. The criteria were verified with the help of questionnaires, which allowed for the determination of the exclusion factors. The research was conducted simultaneously in academic cities in Poland and Vietnam in 2019–2020. The study was conducted via e-mail. The research groups were recruited among volunteers, namely, students in Vietnam (Hanoi) and Poland (Krakow and Warsaw). Information about the possibility of participating in the research was promoted among students. Women who met the criteria for inclusion in the project were asked to invite their friends to participate, i.e., a non-random sampling method (“snowball sampling technique”). Participants were invited by e-mail. The purpose of the study was explained to all participants. People were asked to consent to participate in the research and were informed that participation was voluntary and anonymous. The research plan was to cover 300 Polish and 300 Vietnamese women aged 18–30. In total, the final study included 661 women aged 19–26, with 233 Polish women and 428 Vietnamese women. Due to errors in filling out the questionnaires (incompleteness of the obtained research data) and the participation of people of a nationality other than Polish or Vietnamese, 67 Polish women were excluded from the study and 92 Vietnamese ones. The mean BMI values were 21.02 (Standard Deviaton (SD) = 2.48) for Vietnamese women and 21.53 (SD = 3.67) for Polish women. The mean age of the Polish women was 20.86 (SD = 6.72) and 20.25 (SD = 1.285) years for the Vietnamese women. The women in both groups were students of similar faculties, namely, humanities and social sciences. The respondents were students of large universities who lived in cities that are academic centers in Poland or Vietnam. The similar average age for both groups confirms the intentional similarity of the groups in terms of the variable age.

### 2.2. Procedure

Ethical approval was obtained from the relevant institutional ethical review committees and the research was conducted in accordance with national and international regulations and guidelines. Written consent was obtained from all participants. The protocol of this study was approved by the Ethics Board for Research Projects at the Institute of Applied Psychology, Jagiellonian University in Krakow.

### 2.3. Methods

The Eating Disorders Questionnaire (EDI-3) was used to measure the variable of psychological factors and anti-health (bulimic and restrictive) eating behavior [60]. The Multidimensional Body-Self Relations Questionnaire-Appearance Scales (MBSRQ-AS) [62] was used to measure body image. Additionally, variables such as age, sex, weight, and height were collected.

#### 2.3.1. The Eating Disorders Inventory—EDI-3

The Eating Disorders Inventory (EDI-3) by Garner [60] with the Polish adaptation by Żechowski [73] was used. The Vietnamese translation of EDI-3 scale was developed through a process of translation and independent back-translation. It was made by bilingual psychology researchers—first researcher translated the EDI-3 from English into Vietnamese, and then the other translated the EDI-3 back into English. The differences between the versions were discussed with group of experts psychologists and bilingual translators to ensure translation equivalency of every items. Besides translation, some kinds of cultural adaptation been made, including:Item 3. I wish that I could return to the security of childhood.

Translation: “Tôi ước tôi có thể quay trở lại với sự an toàn của thời thơ ấu”.

However, Vietnamese people don’t use an expression: “sự an toàn của thời thơ ấu”/“the security of childhood”. We discussed it and decided translate this item into understandable and common Vietnamese as below:

Tôi ước tôi có thể quay trở lại với tuổi thơ bình yên (I wish that I could go back to childhood peace).

Item 61. I eat or drink in secrecy.

Translation: Tôi ăn uống trong bí mật.

However, there is not this kind of expression in Vietnam. We discussed it and decided translate this item into understandable and common Vietnamese as below:

Tôi ăn uống mà không để ai biết (I eat and drink without letting anyone know).

The EDI-3 consists of 91 items organized into 12 primary scales, namely, drive for thinness, bulimia, body dissatisfaction, low self-Esteem, personal alienation, interpersonal insecurity, interpersonal alienation, interoceptive deficits, emotional dysregulation, perfectionism, asceticism, and maturity fears. The following three scales from the EDI-3 were used to measure the dependent variable (restrictive and bulimic behaviors): body dissatisfaction (DB), drive for thinness (DT), and bulimia (B). These scales describe eating and body behaviors referred to as eating disorder risk factors [60]. The following satisfactory reliability/validity indices for given subscales were obtained in the presented study: drive for thinness (0.92 in Polish studies and 0.84 in Vietnamese studies), bulimia (0.83 in Polish studies and 0.76 in Vietnamese studies), and body dissatisfaction (0.89 in Polish studies and 0.80 in Vietnamese studies).

The content of items on the DT scale made it possible to measure the strength of the tendency to use behaviors directed at the use of various methods related to the restrictive control of dietary supplements in order to reduce body weight and change one’s own body image. The content of items on the bulimia scale allowed measurement of the strength of the tendency to develop compulsive behaviors such overeating and removing food by inducing vomiting. Due to the DB scale being the third element of the syndrome defined by Garner [60] as risk factors for behaviors typical of eating disorder symptoms, this scale was also included in the analysis of the results of the study when measuring eating behaviors.

Nevertheless, with to the current research model, in the multifaceted definition of the variable mediating body image, the measurement of many different cognitive and emotional aspects of the body image variable was performed using the MBRSQ-AS questionnaire [61,62].

#### 2.3.2. The Multidimensional Body-Self Relations Questionnaire-Appearance Scales

We used the Multidimensional Body-Self Relations Questionnaire-Appearance Scales (MBSRQ-AS) that was developed by Thomas F. Cash et al. [62] with the Polish adaptation by Brytek-Matera and Rogoza [63]. The Vietnamese version of the questionnaire was developed in accordance with the procedure described for EDI-3.

The questionnaire consisted of 34 items assessing the emotional and cognitive image of the individual’s body. The MBRSQ-AS includes 5 scales: self-classified weight (SCW), appearance orientation (AO), appearance evaluation (AE), body area satisfaction (BAS), and overweight preoccupation (OP). Participants evaluated each item of the questionnaire by marking their answers on a five-point Likert scale, ranging from 1 (“definitely disagree”) to 5 (“definitely agree”). The following satisfactory reliability/validity of the MBSRQ indices for given subscales were obtained in the presented study: appearance orientation (0.91 in Polish studies and 0.83 in Vietnamese studies), appearance evaluation (0.87 in Polish studies and 0.82 in Vietnamese studies), body areas satisfaction scale (0.89 in Polish studies and 0.78 in Vietnamese studies), overweight preoccupation (0.79 in Polish studies and 0.70 in Vietnamese studies, self-classified weight (0.66 in Polish studies and 0.53 in Vietnamese studies).

#### 2.3.3. Body Mass Index (BMI)

To measure BMI, participants completed the survey with the following clinical data: age, sex, body mass, and height. BMI was obtained by dividing the body weight in kilograms by the square of the height in meters.

### 2.4. Statistical Methods

Statistical analyses were performed in Statistica 13.3 and Excel (Microsoft Office 365 ProPlus). The stages of statistical analysis included the following:

Stage 1—descriptive statistics, i.e., measuring the mean values of all variables in the research model.

Stage 2—measuring the significance of differences between the average intensity of variables present in the research model between Polish and Vietnamese women. The Mann–Whitney U test was used in this stage of statistical analysis.

Stage 3—measuring the strength of the relationship between the dependent and independent variables using structural equation analysis (path analysis). The aim of this stage was to search for predictors of restrictive and bulimic eating behavior for the Polish and Vietnamese women. The calculations were made with the use of the SPSS AMOS program (Amos (v23.0) (Computer Program). Chicago, IL, USA: IBM SPSS).

In an attempt to explain the direct psychological predictors (among psychological dispositions and features) and the mediating role of the body image in explaining the emergence of restrictive and bulimic behaviors, an integrative model of hypothetical relationships between variables was presented, which was verified by analyzing structural equations. Calculations for the verification of the assumed model of the research variable space were carried out with the use of the SPSS AMOS program. The path model in the exploratory analysis was presented on the basis of the modification of indicators and was reduced (to improve readability) via insignificant paths at the level of *p* > 0.05.

## 3. Results

### 3.1. Characteristics of Psychological Factors, Body Image, and Eating Behavior in Young Polish and Vietnamese Women (Differences between the Groups)

The selection of respondents was deliberate, such that the groups did not differ in terms of age, BMI, or place of study and residence. At the first stage of the statistical analysis of the collected data, the levels of intensity for all variables were measured and the various significances of differences between the Polish and Vietnamese women were measured in relation to all the variables included in the research model. Due to the fact that the studied variables did not have a normal distribution, the non-parametric Mann–Whitney U test was used for the statistical analysis (Table 1).

The results of the statistical analysis are presented in Table 1. The analysis allowed the authors to answer the first research question and indicate the differences between Polish and Vietnamese women in terms of the intensity of various psychological factors (traits and dispositions) and factors describing body image, as well as restrictive and bulimic behaviors. A comparative analysis of the mean values of all variables in the Polish and Vietnam groups showed the presence of significant differences between these groups in terms of certain variables:The average BMI in research groups was similar, the Vietnamese women turned out to not differ significantly (mean (M) = 21.02) from the Polish women (M = 21.53) of the same age (*p* > 0.05). The BMI values were within the normal weight ranges for the age of life in both trials.With the analyzed psychological factors, only interpersonal alienation and striving for thinness did not differentiate between Polish and Vietnamese women. All other factors (dispositions and psychological features) turned out to be significantly different between Polish and Vietnamese women. Vietnamese women obtained significantly higher mean severity values in terms of bulimic behavior and significantly higher values in relation to Polish women for low self-esteem, sense of alienation, personal alienation, lack of self-confidence (interpersonal insecurity) and interpersonal alienation. The differences were significantly higher (the highest level of difference between respondents) in terms of the intensity of emotional dysregulation, interoceptive deficits, perfectionism, asceticism, and the fear of maturity.The comparative analysis of the mean values also showed significant differences between Polish and Vietnamese women in terms of the body image. Vietnamese women obtained a significantly lower level of perceived satisfaction with their overall appearance and lower satisfaction with individual body areas than Polish women. Vietnamese women were also significantly less oriented towards looking after their appearance than Polish women (appearance orientation). Vietnamese women also showed a significantly lower level of preoccupation towards being overweight and the fear of obesity (overweight preoccupation) than Polish women, and the frequency of focusing and monitoring their own weight, using various diets and dieting overall, and lower levels of self-classified weight were all lower than Polish women; however, although the indicated differences were statistically significant, they were lower than the differences in the case of other psychological factors included in the study profile. Polish women were more focused on the fear of gaining weight and body weight but also show a higher level of satisfaction with their own body and care for appearance than Vietnamese women.

### 3.2. Psychological Predictors of Restrictive and Bulimic Eating Behavior

On the basis of in-depth causal inference with multidirectional and multivariate analyzes, the main goal of the study was to establish a model of major psychological predictors influencing the development of anti-health (restrictive and bulimic) eating behaviors, which is defined as the explanatory variable here. With the prognosis of high indicators of goodness of fit for the model, based on empirical data, it can be concluded that it represents the correlation matrix well. Figure 2 present indicators such as CHI^2^ (chi square goodness of fit), RMSEA (Root mean square error of approximation), CFI (Comparative Fit Index), and the Hoelter coefficient, and these prove the good fit of the model to the empirical data. The CHI^2^ and RMSEA indices show a good level of fit and the Goodness of fit index (GFI index) shows a very good level of fit. The developed model of the paths for the entire group of Polish women (*n* = 233) and Vietnamese (*n* = 428) was verified in terms of matching the raw data. The fit indicators obtained at this stage suggested a good fit for the constructed model. The applied analysis of structural equations allowed the authors to answer research questions concerning the search for predictors of restrictive and bulimic eating behavior in Polish and Vietnamese women, as well as defining the direct and indirect influences of factors describing body image on the development of restrictive and bulimic behaviors towards eating for young Polish and Vietnamese women. Intercultural differences between Polish and Vietnamese women were also verified in terms of the verified variables of the research and the predictors of restrictive and bulimic eating behaviors.

#### 3.2.1. Analysis of Psychological Predictors of Restrictive and Bulimic Eating Behaviors in a Group of Polish Women

Figure 2 presents a path model for the sample of Polish women and the values of path coefficients showing the degrees of influence and thus explaining psychological factors for the development of restrictive and bulimic behaviors towards eating for Polish women.

By analyzing the significant paths and path coefficients presented in Figure 2, for the group of Polish women, it can be indicated that the strongest direct causal effects explaining the development of restrictive behaviors and the pursuit of thinness in Polish women turned out to be the following psychological features and dispositions:Low self-esteem (0.305; *p* = 0.001): The lower the self-esteem, the greater the strive for thinness and the prevalence of restrictive behavior in Polish women.Asceticism (0.265; *p* = 0.001): The greater the intensity of asceticism, the greater the strive for thinness and restrictive behavior in Polish women.Interoceptive deficits (0.233; *p* = 0.001): The higher the interoceptive deficits, the greater the intensity of striving for thinness and the prevalence of restrictive behavior in Polish women.

As for the influence of the mediating variable, i.e., body image, on the restrictive behavior undertaken and the pursuit of thinness in Polish women, the strongest significant path effects were:Overweight preoccupation—preoccupation with being overweight and assessment of the level of fear of obesity (0.153; *p* = 0.001). The greater the fear of obesity and preoccupation with being overweight, the greater its influence on restrictive behavior and striving for thinness in Polish women.Appearance orientation (0.099; *p* = 0.037): The greater the attention to appearance, the greater the strive for thinness and restrictive behavior in Polish women.

The strongest total causal effect explaining the influence on the frequency of bulimic behavior (using various body cleansing agents) is explained by two variables that are described by the profile of psychological traits of Polish women:Asceticism (0.356; *p* = 0.001): The greater the intensity of asceticism, the greater the intensity of bulimic behavior in Polish women.Interoceptive deficits (0.325; *p* = 0.001): The greater the interoceptive deficits, the greater the prevalence of bulimic behavior.

When it comes to the influence of the mediating variable, i.e., body image, on bulimic behavior in Polish women, only overweight preoccupation and the assessment of the level of fear of obesity (−0.132; *p* = 0.033) were relevant. The greater the anxiety against obesity and preoccupation with overweight, the smaller its impact on bulimic behavior in Polish women. 

On the other hand, based on the obtained values of the path coefficients, the strongest total causal effect, explaining the influence of only selected aspects of the variable body image, showed the following psychological dispositions and features (the explanatory variable):Interpersonal alienation showed a significant impact on both appearance evaluation (0.287; *p* = 0.002) and body area satisfaction (0.272; *p* = 0.004). The greater the level of interpersonal alienation for Polish women, the greater the impact on general appearance satisfaction and BAS.Personal alienation showed a significant negative impact on appearance evaluation, where the greater the personal alienation, the smaller the impact on the assessment of satisfaction with appearance (−0.230, *p* = 0.028). Similarly, maturity fears showed a significant negative direct impact on appearance evaluation (−0.210; *p* = 0.031), where the greater the level of fear of adulthood, the lower the level of body appearance satisfaction for Polish women.Interoceptive deficits showed a significant direct impact on body area satisfaction (−0.231; *p* = 0.026), where the greater the interoceptive deficits were in Polish women, the lower the level of satisfaction with the appearance of individual body areas, and the same psychological variable showed a direct impact on overweight preoccupation (0.280, *p* = 0.007), where the greater the interoceptive deficits, the greater the impact of the level of anxiety about gaining weight and the frequency of monitoring one’s own weight (weight vigilance).

The strongest total causal effect explaining the effect on appearance orientation was simultaneously demonstrated by the following dispositions and psychological characteristics:Low self-esteem (−0.332; *p* = 0.007): When self-esteem is lower and the sense of worthlessness is higher, the level of care for appearance is lower in Polish women.Interoceptive deficits (0.236; *p* = 0.021): The higher the interoceptive deficits, the higher the impact on the higher level of care for appearance in Polish women.Perfectionism (0.176; *p* = 0.016): With higher perfectionism, the greater its importance becomes for caring for appearance and the development of striving for thinness and restrictive behavior, but with a lower intensity of bulimic behavior.

#### 3.2.2. Analysis of Psychological Predictors of Restrictive and Bulimic Eating Behavior in the Group of Vietnamese Women

Models of significant pathways and path coefficients describing psychological predictors of restrictive and bulimic behavior in Vietnamese women and the role of the body image in the formation of these behaviors are presented in Figure 3.

By analyzing the models of significant paths and path coefficients presented in Figure 3, it can be indicated that the strongest direct causal effects explaining the influence on the development of restrictive behavior and focusing on the pursuit of thinness in Vietnamese women turned out to be the following psychological features and dispositions:Asceticism (0.123; *p* = 003): The higher the level of asceticism, the greater its influence on undertaking restrictive behavior and striving for thinness.Interoceptive deficits (0.101; *p* = 0.018): The higher the deficits in interoceptive awareness, the greater their direct impacts on developing restrictive behaviors and striving for thinness.Personal alienation (−0.096; *p* = 0.032): The greater the level of personal alienation, the smaller its direct impact is on the pursuit of thinness and restrictive behavior for Vietnamese women.

As for the influence of the mediating variable, i.e., body image, on restrictive behavior and the pursuit of thinness in Vietnamese women, the most significant path effects were:Overweight preoccupation, i.e., the assessment of the level of fear of obesity (0.808; *p* = 0.001), where the greater the fear of obesity and preoccupation with being overweight, the greater its impact on restrictive behavior and thinness in Vietnamese women. Overweight preoccupation represented the strongest impact in the entire path model.Body area satisfaction, i.e., the assessment of satisfaction with individual body areas (−0.384; *p* = 0.001), where the better the assessment of satisfaction with individual areas of the body, the smaller their impact on restrictive behavior and striving for thinness in Vietnamese women.Appearance orientation (−0.062; *p* = 0.031) and appearance evaluation (0.068; *p* = 0.047), where the greater the focus on looking after the appearance, the lower the restrictive behavior, but also the greater the satisfaction with the assessment of the appearance and stronger restrictive behavior development and focus on the pursuit of body thinness.

The strongest overall causal effect explaining the impact on the frequency of bulimic behavior (using various body cleansing agents) was represented by two variables described by the profile of psychological traits of Vietnamese women:Interoceptive deficits (0.328, *p* = 0.001): The greater the interoceptive deficits, the greater the tendency to develop bulimic behavior.Ascetism (0.223; *p* = 0.001): The greater the ascetism, the greater the tendency to develop bulimic behavior.

As for the influence of the mediating variable, i.e., body image, on bulimic behavior in Vietnamese women, the strongest significant pathway effects were:Overweight preoccupation (0.117; *p* = 0.009): The greater the preoccupation with being overweight and the fear of obesity, the greater the intensity of bulimic behavior.Appearance evaluation (0.014; *p* = 0.033): The higher the satisfaction with the appearance of the body, the greater the intensity of bulimic behavior.

Moreover, it was shown that low self-esteem directly influences the assessment of satisfaction with appearance (−0.483, *p* = 0.001) and satisfaction with individual body areas (−0.147; *p* = 0.001). A greater feeling of low value represents a lower level of satisfaction with general appearance and individual body areas. Here, a further significant (though slight) mediating positive influence of satisfaction with the general appearance on the pursuit of thinness and restrictive behavior was demonstrated. The assessment of satisfaction with individual areas of the body mediates low self-esteem for Vietnamese women and has a similar and negative impact on restrictive behavior and striving for thinness. The higher the satisfaction with individual areas of the body, the lower the restrictive behavior.

In summary, in the Vietnamese research model, asceticism and interoceptive deficits showed the strongest direct influence on restrictive and bulimic behaviors, where the greater the level of asceticism and interoceptive deficit, the greater the intensity of developing risky behaviors for restrictive and bulimic eating habits. On the other hand, for body image, satisfaction with the appearance of the body and the fear of gaining weight are the variables that mediate restrictive and bulimic behaviors. Low self-esteem and asceticism are psychological traits that show a significant direct impact on the self-esteem pertaining to body appearance and the fear of gaining weight. Low self-esteem affected body image but did not show a significant direct effect on restrictive and bulimic behavior in Vietnamese women, while, on the other hand, asceticism showed a direct influence on both restrictive and bulimic behaviors, and, at the same time, a direct influence on satisfaction with appearance and the fear of gaining weight (overweight preoccupation).

## 4. Discussion

### 4.1. The Characteristics of Psychological Factors as Well as Restrictive and Bulimic Behaviors in Polish and Vietnamese Women

The results of the study indicate that Vietnamese women show a significantly higher intensity in the use of bulimic behaviors than Polish women and significantly lower self-esteem than Polish women, along with a higher sense of personal alienation, higher lack of self-confidence (interpersonal insecurity) and interpersonal alienation, and significantly higher levels of emotional dysregulation, interoceptive deficits, perfectionism, asceticism, and fear of maturity. There were also no differences between Polish and Vietnamese women in terms of the intensification of restrictive behaviors and the pursuit of thinness. These results have been confirmed by other studies describing the importance of body dissatisfaction for the development of typical behaviors in eating disorders, such as in the Swedish studies by Forsén Mantilla and Birgegard [74], the Polish studies by Izydorczyk et al. [32,75], and the Brazilian studies by de Carvalho, Alvarenga and Ferreira [49]. On the other hand, Rodgers and Melioli [51], in a review of 67 studies (emphasizing the important role of social media (i.e., the internet) and not taking into account psychological factors in these studies), indicated a significant relationship between the body image and behaviors related to the development of eating disorders in many countries in Europe, America, Australia, and Asia. The same situation was found in a review of 20 peer-reviewed studies by Holland and Tiggemann [50], where the authors of the review pointed to a relationship between the influence of social media on body appearance and behavior typical for the development of eating disorders; however, Holland and Tiggemann [50] did not indicate studies describing the relationship between the multifactorial model of psychological traits and the development of restrictive and bulimic behaviors typical for eating disorders. Due to the lack of comparative studies identified in the literature by the authors of this article, describing the relationship between the multifactorial model of psychological variables (psychological features and dispositions, bulimic compulsive and restrictive behaviors, and body image) in populations of women from Europe and Asia (also Vietnamese and Polish women), it is difficult to compare these results with the results of other studies. On the other hand, the authors of this article, comparing the results of their study, referred to the results of the Vietnamese study by Ko et al. [17] and other results of the study by Izydorczyk and others concerning the comparison of Japanese and Polish women in terms of the profiles of psychological characteristics and dispositions, as well as the specificity of the body image [32]. Research by Ko et al. [17] was conducted on a population of 244 Vietnamese female students using eating disorder questionnaires and showed the dependencies between BMI and compulsive and restrictive eating behaviors in the examined Vietnamese women. The mean result for underweight Vietnamese women (M = 14.79, SD = 6.81) indicated a lower level on the scale of striving for thinness in the EDI-2 questionnaire when compared to people with a normal body weight (M = 24.65, SD = 6.86) and those who were overweight (M = 31.33; SD = 6.66). Vietnamese women who were underweight (M = 27.24, SD = 7.57) were less dissatisfied with their body than those with normal body weight (M = 35.94, SD = 8.67) and those who were overweight (M = 43.33, SD = 11.24). The results of the study by Ko et al. [17] acknowledge the growing desire for thinness and body dissatisfaction among young women in Vietnam. In the studies by Izydorczyk and their team, the BMI values of Vietnamese and Polish women were within normal ranges for age and life. Due to the objectives of the study, the relationship between BMI and the intensities of restrictive, compulsive, and bulimic behaviors were not analyzed. On the other hand, Saleh et al. [76], conducting a study using the Eating Attitudes Test (EAT 26) on a population of 2001 students at the An-Najah National University in Palestine, indicated a significant positive relationship between BMI and restrictive and bulimic behaviors. The surveyed female students obtained better results than female students of science and medicine. Saleh et al. [76] did not compare Palestinian women with women of other nationalities, and age was significantly and negatively correlated with the results of the EAT-26 test in terms of describing restrictive and bulimic eating behaviors. As indicated by Izydorczyk et al. [32], in studies conducted on Polish and Japanese women, the latter (similarly to the Vietnamese women in the research by Izydorczyk and the team presented in this article) showed a higher level of compulsive and bulimic behaviors than Polish women. Moreover, the Vietnamese women showed a similar level of striving for thinness and related restrictive behaviors as Polish and Japanese women, as well as a similar maturity fears [32]. Vietnamese and Japanese women with a similar age also showed a significantly higher level of emotional dysregulation than the two different groups of Polish women who were of a similar age [32]. Vietnamese women showed a significantly higher level of asceticism and perfectionism than Polish women, while, in comparison to other studies by Izydorczyk and their team, the same age group of Polish women showed a significantly higher level of asceticism and perfectionism than the group of Japanese women. In the studies of Izydorczyk and their team, Japanese women showed no differences with Polish women in terms of the intensities of interoceptive deficits, low self-esteem, personal alienation, higher lack of self-confidence (interpersonal insecurity), and interpersonal alienation [32]; however, in comparative studies of authors cited in this article, a group of Polish women studied with similar age showed significantly lower intensities of interoceptive deficits, low self-esteem, personal alienation, and a higher lack of self-confidence (interpersonal insecurity) and interpersonal alienation (Table 1). The Polish women surveyed by Izydorczyk et al. showed less dissatisfaction with their body than the Vietnamese women. In turn, the surveyed Polish women showed no differences in terms of body dissatisfaction compared to the Japanese women [32]. As shown by other studies, Japanese women tend to have restrictive and bulimic behaviors while striving for a perfect body appearance [35,66]. The study results by the authors of this article also indicate that Vietnamese women, compared to Polish women, are characterized by a higher level of emotional dysregulation and higher importance in that regard in the development of eating disorders. In the study by Chapuis-de-Andrade, et al. [77], conducted online on a population of 27,501 Brazilian respondents aged 18 to 55 (including 69.6% of women with a normal BMI, mostly Caucasian, aged M = 28.9; SD = 8.7 years), their results indicated a significant relationship between overly irritable cyclothymic affective traits and compensatory eating behaviors. The Chapuis-de-Andrade study [77] also did not make cross-cultural comparisons and did not examine the psychological variables included in the study model of the authors of this article. The discussion of the results of intercultural comparisons between Polish (European culture) and Vietnamese (Asian culture) women in terms of the characteristics of significant psychological predictors of restrictive and bulimic behaviors is presented below.

### 4.2. Psychological Predictors of Behavior towards Eating with a Polish and Vietnamese Comparison

#### 4.2.1. Interoceptive Deficits, Asceticism, and Low Self-Esteem as Predictors of Restrictive and Bulimic Behaviors

The study results obtained by the authors of this article confirm that asceticism and interoceptive deficits are the most important predictors of the development of restrictive and bulimic behaviors in the sample groups of Polish and Vietnamese women. The greater the severity of asceticism and interoceptive deficits, the greater the severity of restrictive and bulimic eating behaviors in the sampled Polish and Vietnamese women. Obeid et al. [69] similarly indicated the significance of a positive relationship between asceticism, perfectionism, and bulimia in a group of 178 adolescents (91% of girls M = 15.73 SD = 1.31). The results of Obeid et al. [69] indicated that adolescents with symptoms of paroxysmal purging behaviors had higher levels of perfectionism, asceticism, and over-control when compared to subjects with symptoms of restrictive behavior. Higher levels of perfectionism, asceticism, and over-control were associated with increased symptoms of depression and anxiety. The results of the study by Izydorczyk et al. presented in this article confirmed the importance of perfectionism as a predictor explaining the self-esteem relating to body image (appearance evaluation) and the mediating role of perfectionism in the development of restrictive and bulimic behaviors in the group of Polish women; however, in the case of the Vietnamese women, perfectionism was irrelevant in explaining the image of the body and restrictive and bulimic behavior. In the systematic review of studies presented in the literature up to January 2019, presented by Vacca et al. [68], emerging studies suggest a significant role of perfectionism in development for eating disorders. The second significant psychological predictor of restrictive and bulimic behaviors in both Polish and Vietnamese women, shown in the study of the authors of this article, was interoceptive deficits. A metanalysis by Jenkinson et al. [70], including a study with a sample size of 345, also indicates a similarly significant influence of interoceptive deficits in the development of restrictive and bulimic behaviors in eating disorders. Smith et al. [71] and Martin et al. [72] have indicated the significant importance of interoceptive deficits in the development of eating disorders. In the research of the authors of this article, low self-esteem is another psychological predictor, but only for restrictive behaviors and only in the group of Polish women. Low self-esteem has also been indicated to be significant for the development of restrictive and bulimic behaviors in other studies [74,78,79]. Borda Más et al. [78], using the EDI questionnaire and measuring the mediating role of self-esteem and perfectionism in the relationship between personality traits and body dissatisfaction, on a population of 155 women (aged 18 to 31), indicated the significant roles of self-esteem and perfectionism as the main variables mediating the influence of personality traits on body dissatisfaction and restrictive and bulimic behaviors. On the other hand, Forsén Mantilla and Birgegard [74], in a study of 388 healthy women aged 16 to 25 years, showed a relationship between low self-esteem and increasing symptoms of eating disorders (dissatisfaction with the body, restrictive behaviors). Further studies by Forsén Mantilla et al. [74] conducted with 457 women with anorexia and 861 with bulimia, and 505 women with binge eating episodes also confirmed the importance of low self-esteem for the development of eating disorder symptoms. Similarly, in the studies of Mora et al. [80] on a population of 500 Spanish women, low self-esteem significantly increased the probability of obtaining a high score on a scale assessing the level of behavior typical for eating disorders. In the research of the authors of this article, emotional dysregulation turned out to be significantly higher in the population of women from Vietnam than in the population of Polish women. Nevertheless, emotional dysregulation did not turn out to be a predictor of restrictive and bulimic behavior, neither in the group of Polish women nor in the group of Vietnamese women. This result may be related to the specifics of the sample of the studied women and the use of different tools for measuring this psychological variable. The research review and meta-analysis of Prefit et al. [67] indicates a significant relationship between emotional regulation and behaviors characteristic of eating disorders. Agüera et al. [58], conducting research on, among others, a population of 286 Spanish women and 78 without eating disorders showed a correlation between emotional dysregulation and a dysfunctional personality profile and the intensity of compulsive and bulimic behaviors typical of eating disorders. As mentioned already in Section 4.1., the study conducted by Chapuis-de-Andrade et al. [77] on a sample of 27,501 volunteers aged 18 to 55 years (M = 28.9; SD = 8.7 years, 69.6% of which are Brazilian women) indicated a relationship between temperamental factors and compensatory (bulimic) behaviors. On the other hand, a study by Mallorquí-Bagué et al. [81] also indicated the presence of difficulties in emotional regulation in a group of 69 people with eating disorders. Personal alienation turned out to be the last of the variables distinguished as a predictor (albeit of low significance) only for restrictive behavior in Vietnamese women. The higher the intensity of personal alienation, the lower the level of restrictive behavior in the Vietnamese women. No such relationship was found in Polish women. Christensen and Haynos [82], Levine [83], and Hempel et al. [84] have also indicated the importance of the sense of loneliness and negative interpersonal relationships as risk factors for eating disorders (i.e., anorexia, bulimia, and compulsive overeating).

#### 4.2.2. Body Image as an Intermediary Variable between Psychological Factors and Restrictive and Bulimic Behaviors.

The study results obtained by the authors of this article indicate that only the overweight preoccupation variable (body image component) turned out to be a common predictor of restrictive and bulimic behaviors for Polish and Vietnamese women. In turn, appearance orientation was the second common predictor for Polish and Vietnamese women, while appearance evaluation was only a predictor for restrictive and bulimic behaviors for Vietnamese women. Body area satisfaction was only a predictor for restrictive behavior in Vietnamese women. These relationships were not found in Polish women. As for the significant relationship between the components describing the variable of body image (describing self-esteem and body satisfaction) and compulsive and bulimic behaviors, this has also been confirmed by studies on the population of Brazilian women published by de Carvalho and others [49]. These studies included a sample of 741 Brazilian female and male students with an average age of 23.55 (SD = 4.09). The results for the Brazilian sample, referring to the indication of the importance of social factors, confirm the existence of a significant correlation between the increase in dissatisfaction with the body and a restrictive eating behavior [49]. On the other hand, low self-esteem turned out to be a predictor of appearance evaluation and body area satisfaction for both Polish and Vietnamese women. Barker and Bornstein [85], in their research with a sample of 130 American teenagers, also indicated a significant correlation between global self-esteem and the self-esteem of the body and satisfaction with appearance. Similarly, O’Dea [86] indicated that people with lower self-esteem are more likely to report dissatisfaction with the size or shape of their body, regardless of age or gender. In the studies by Izydorczyk, et al. [87], the self-esteem of 95 healthy Polish teenagers also showed a correlation with body image. In the EDI-3 questionnaire presented by Garner [60] in their handbook, the research also indicates the importance of low self-esteem in the development of body dissatisfaction. The study results obtained by the authors of this article also indicate that body self-esteem is described by appearance evaluation, appearance orientation and body area satisfaction, and that these variables are significantly (albeit to a low degree) explained for the group of Polish women by interpersonal alienation, personal alienation, perfectionism, maturity fears, and interoceptive deficits. In the group of Vietnamese women, there was no such relationship. No comparative studies of Polish women with other populations that describe the above-mentioned selected relationships between the sense of loneliness and the fear of gaining weight have been identified, hence it is difficult to provide a comparative analysis of this aspect. On the other hand, the relationship between the fear of weight gain, perfectionism, and interoceptive deficits, and body image satisfaction was presented in Section 4.2.1. where the relationship between perfectionism and interoceptive deficits and restrictive and bulimic behaviors was indicated.

### 4.3. Limitations of the Study

The presented research represents a rare and even unique opportunity to provide cultural comparisons between women of two different nationalities that have been brought up in different cultures (Asian and European); however, the results of this study are also subject to some limitations. The groups of Polish and Vietnamese women surveyed, although sufficient in terms of numbers for statistical analysis, were not very large. Another limitation is the smaller number of Polish women than the group of Vietnamese women, which resulted from unforeseen difficulties in the process of collecting research data during their conduct and the need to complete them; however, statistical analysis made it possible to make comparisons that have previously been missing in the literature between two clearly different cultures and attitudes towards nutrition (i.e., “rice culture” vs. “potato culture”) and attitudes towards the body for women brought up in an Asian culture and a Western culture. It should also be noted that the selection of the group of respondents was difficult due to the geographical location and access to the respondents. Due to the pandemic situation (COVID-19), the research has been completed online. Due to BMI status was estimated based on self-reported height and weight. However, this method of measurement is accepted to calculate BMI for weight classification purposes. Over the past months, the use of online and mobile communication, including research methods like online surveys has soared. The distance online learning have been implemented in Poland and Vietnam, and the Internet access is available for young people. There is evidence that online research design may facilitate cross-cultural research, and generate big sample sizes. Statistical analysis showed that the distribution of the examined variables did not meet the standards of a normal distribution. Although the purpose of this study was to identify specific intercultural differences, it should be emphasized that both groups featured a similar narrow age range. Extending this study to more distant populations would lead to greater differentiated results in terms of differences and similarities between the studied groups regarding predictors of bulimic and restrictive behavior. The research requires further deepening on subsequent populations of women of different ages.

## 5. Conclusions

Intercultural differences between the studied groups of Vietnamese and Polish women show that Vietnamese women show greater intensities for all psychological variables verified in the research model (i.e., low self-esteem, personal alienation, interpersonal insecurity, interpersonal alienation, emotional dysregulation, interoceptive deficits, perfectionism and asceticism, and anxiety) before adolescence, apart from interpersonal alienation and the pursuit of thinness, the levels of which are similar to Polish women. As for body image, the Vietnamese group achieved a significantly lower level than the Polish group for the perceived satisfaction with their general appearance (appearance evaluation) and lower satisfaction with particular body areas. Vietnamese women were also significantly weaker than Polish women in terms of appearance orientation. Vietnamese women also showed a significantly lower level of preoccupation with being overweight and fear of obesity than Polish women (overweight preoccupation) and the frequency of focusing and monitoring one’s own weight, the frequency of using various diets and weight loss, and a lower level of focus on body weight than Polish women (self-classified weight). The study analysis allowed for the identification of common predictors of restrictive and bulimic behaviors and psychological predictors explaining body image for the surveyed Polish and Vietnamese women. These predictors were considered to be psychological risk factors for the development of restrictive and bulimic eating behavior. They are presented collectively in Table 2.

In summary, it is necessary to point out the practical implications concerning the need to diagnose psychological predictors (risk factors) of eating disorders. The measurement of these factors may be necessary in the practice of family doctors, psychiatrists, pediatricians, educators, and other specialists who support health prophylaxis in women (especially in terms of shaping a positive body image and preventing eating disorders). The global westernization in emerging countries seems to be a fact. However, there are some cultural conditions that may influence on how women experience themselves. Both, Vietnamese and Polish women seem to be focused on appearance. However, negative a critical and pejorative attitude towards oneself seems to be crucial in the group of young Vietnamese women. Polish women, by contrast, seem to be rather focused on efforts to not gain the weight. In line with health prophylaxis, an in-depth discussion on the risk factors of eating disorders and the body image standards of women brought up in different cultures should be present in the creation of preventive programs for anti-health eating behaviors (restrictive and bulimic) and body image for adolescents, young women, and other adults struggling with different eating disorders.

## Figures and Tables

**Figure 1 nutrients-13-00910-f001:**
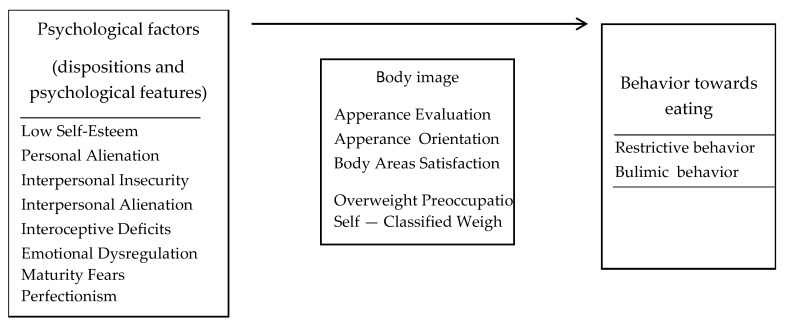
Theoretical model of the research variables (own elaboration).

**Figure 2 nutrients-13-00910-f002:**
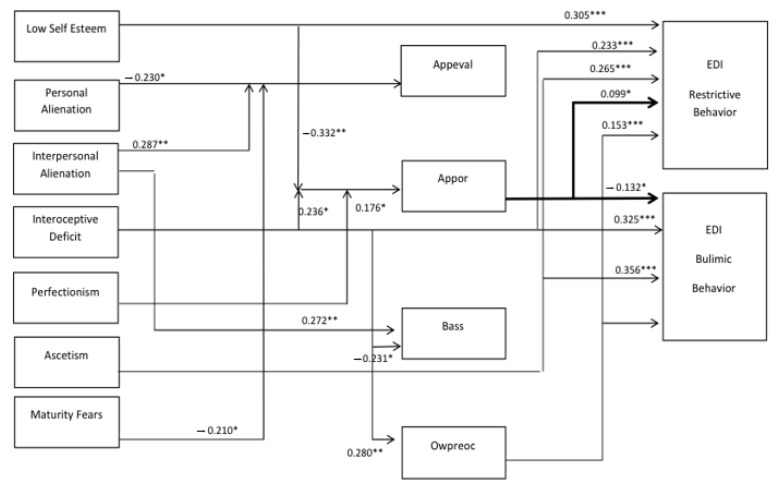
Pathway model (SEM) of factors explaining the development of restrictive and compensatory eating behaviors in the group of young Polish women (*n* = 233). For chart readability, only the important paths are presented and the graphic markings of residues are omitted. (CHI = 1.95; *p* > 0.05; RMSEA = 0.060; CFI = 0.986;) * *p* < 0.05; ** *p* < 0.01; *** *p* < 0.001 (significant for two-sided test). Appearance evaluation (Appeval); appearance orientation (Appor); body area satisfaction (BAS); overweight preoccupation (Owpreoc). CHI^2^ (chi square goodness of fit), RMSEA (Root mean square error of approximation), CFI: chi square goodness of fit, EDI: Eating Disorder Inventory.

**Figure 3 nutrients-13-00910-f003:**
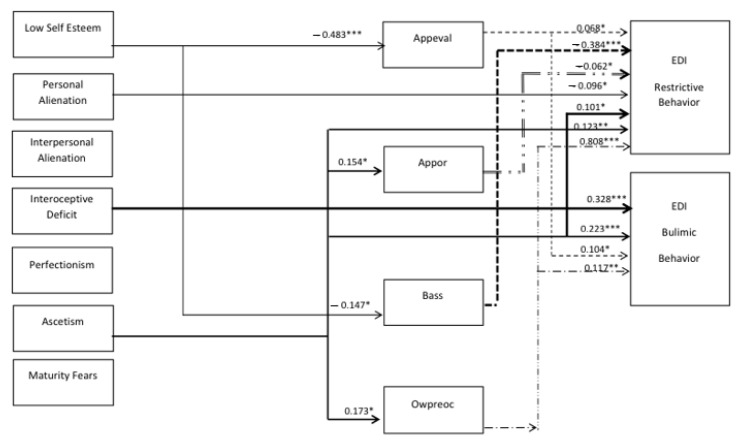
Pathway model, Structural Equation Modeling (SEM) of factors explaining the development of restrictive and compensatory eating behaviors in the group of young Vietnamese women (*n* = 428). For the readability of the chart, only the important paths are presented and the graphic markings of residues are omitted. (CHI = 1.56; *p* > 0.05; RMSEA = 0.037; CFI = 0.992;) * *p* < 0.05; ** *p* < 0.01; *** *p* < 0.001 (significant for the two-sided test).

**Table 1 nutrients-13-00910-t001:** Comparative analysis between groups of Vietnam (*n* = 428) and Polish (*n* = 233) women in terms of the variables included in the research model. Mann-Whitney U test.

Research Variable	Polish Women	Vietnamese Women			
	Me (25Q-75Q)	Me (25Q-75Q)	U	Z	*p*
Eating Disorders Questionnaire (EDI-3) Behavior towards eating					
Driver for thinnes	10.00(5.00–17.00)	10.00 (5.00–17.00)	79,513.0	0.34	0.730
Bulimia	3.00 (1.00–7.00)	6.00 (3.00–10.00)	56,919.5	−7.13	0.001
Body Dissatisfaction	18.00 (14.00–21.00)	22.00 (18.00–26.00)	51,355.0	−8.80	0.001
Psychological factors					
Low Self-Esteem	12.00 (9.00–15.00)	13.00 (10.00–16.00)	67,916.5	−3.83	0.001
Personal Alienation	12.00 (8.00–15.00)	14.00 (12.00–17.00)	56,097.0	−7.38	0.001
Interpersonal Insecurity	16.00 (11.00–20.00)	18.00 (15.00–20.00)	64,667.5	−4.81	0.001
Interpersonal Alienation	12.50 (8.00–16.00)	12.00 (11.00–15.00)	76,013.0	−1.40	0.162
Interoceptive Deficits	11.00 (6.00–17.00)	14.00 (10.00–19.00)	60,302.0	−6.12	0.001
EmotionalDysregulation	8.00 (4.00–12.00)	13.00 (9.00–17.00)	48,208.0	−9.75	0.001
Perfectionism	8.00 (4.00–12.00)	11.00 (8.00–14.00)	58,282.0	−6.72	0.001
Asceticism	6.00 (4.00–10.00)	8.00 (5.00–11.00)	61,434.0	−5.78	0.001
Maturity Fears	13.00 (9.00–16.00)	19.00 (16.00–23.00)	29,504.5	−15.37	0.001
Multidimensional Body-Self Relations Questionnaire (MBSRQ-AS)					
Apperance Evaluation	3.14 (2.71–3.71)	3.00 (2.57–3.43)	47,362.0	4.12	0.001
Apperance Orientation	3.50 (3.17–3.83)	3.25 (3.00–3.42)	37,703.5	7.73	0.001
Body Areas Satisfaction	3.44 (2.89–3.78)	2.78 (2.22–3.33)	33,193.5	9.41	0.001
Overweight Preoccupation	2.50 (2.00–3.00)	2.00 (1.50–3.25)	49,239.5	3.41	0.001
Self-Clasiffied Weigh	3.00 (3.00–3.50)	3.00 (2.00–3.50)	48,466.5	3.70	0.001

U: Mann Whitney U test; Z: Z score; *p*: Significant level.

**Table 2 nutrients-13-00910-t002:** Summary of psychological risk factors (predictors of restrictive and bulimic behaviors) common and differentiating for the group of Polish and Vietnamese women.

Explained Variables Psychological Factors	Predictors Group of Vietnamese Women	Predictors Group of Polish Women
Apperance evaluation	Low Self Esteem β = −0.483 ***	Personal Alienation β = −0.230 *
Apperance Orientation	Ascetism β = 0.154 *	Low Self Esteem β = −0.332 ** Interoceptive Deficit β = 0.236 * Perfectionism β = 0.176 **
Βody Areas Satisfaction	Low Self Esteem β = −0.147 *	Interpersonal Alienation β = 0.272 ** Interoceptive Deficit β = −0.231 *
Overweight Preoccupation	Ascetism β = 0.173 *	Interoceptive Deficit β = 0.280 **
Restrictive Behaviours	Apperance Evaluation β = 0.068 *Body Areas Satisfaction β = −0.384 *** Apperance Orientation β = −0.062 * Personal Alienation β = −0.096 * Interoceptive Deficit β = 0.101 * Ascetism β = 0.123 ** Overweight Preoccupation B = 0.808 ***	Low Self Esteem β = 0.305 *** Interoceptive Deficit β = 0.233 *** Ascetism β = 0.265 *** Apperance Orientation β = 0.099 *Overweight Preoccupation β = 0.153 ***
Bulimic Behaviours	Apperance Evaluation β = 0.104 * Interoceptive Deficit β = 0.328 *** Ascetism β = 0.223 *** Overweight Preoccupation β = 0.117 **	Appor β = −0.132 * Interoceptive Deficit β = 0.325 *** Ascetism β = 0.356 ** Overweight Preoccupation β = −0.132 *

* *p* < 0.05, ** *p* < 0.01, *** *p* < 0.001.

## Data Availability

The data of this study are available from the corresponding author upon request.

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
