# Peer review of "Psychological Risk Factors for the Development of Restrictive and Bulimic Eating Behaviors: A Polish and Vietnamese Comparison"

_nutrients, 2021, doi:10.3390/nu13030910_

Round 1

Reviewer 1 Report

The paper is about an interesting topic: the comparison between two different socio-cultural populations as regards eating psychopathology. I think the paper is well written and the design correct.

I am only concern about the different numerosity of the samples.

Polish people are half than Vietnamese people. This could generate a statistical error for the "weight" of the bigger one. Did the authors consider this point?

Moreover, the use of an online survey could have an impact on the economic status of the included sample? Maybe with more "western-style" people? 

Author Response

We would like to thank the Reviewer for the further comments on our manuscript. We proceeded with the revision of the manuscript taking into account all the comments. Below  we explain in detail how we addressed those issues.

Reviewer 2 Report

1. Some issues regarding English language and terminology should be corrected:

  • As examples of missuse of some words: replace "This aim" with "The aim" (line 14), the use of "where" (end of line 321);
  • The use of "universal predictors" (e.g. lines 14, 69, 110, 119) should be replaced with a more adequate designation, as only two countries are being compared;
  • The use of the term "significant(ly)" should be avoided when not referring to statistical significance (e.g. line 40);
  • Also, some terms indicating cause-effect relationships (e.g. "influence" in line 181) shouldn't be used considering that the study has a cross-sectional design;
  • In line 91, the use of "personalities" is innacurate; the use of "personality traits" (or similar) should be considered.

2. In sections 1.1 and 1.2 the authors should focus on the level of industrialization and/or Westernization (as they do in lines 60-62) and cultural differences, and not in the idea of communist/ post-communist countries or periods. Please note that several currently communist countries have high levels of industrialization and, regarding some issues, even Westernization. 

3. If available, please provide data on how many potential participants were contacted in each country. If this data is available, the authors should compare the two samples regarding participation rate, using an adequate hypothesis test.

  • Moreover, please confirm that only data from the Polish sub-sample were excluded (lines 210-211).
  • These issues should be addressed in the limitations, and not only referring the "difficulties in the process of collecting research data" (lines 727-728), since the main limitation is not the "smaller number of Polish women", but the potential selection bias.

4. The authors report that "the studied variables did not have a normal distribution" (line 311), but still they describe them using means and standard deviations, which should be corrected. Also, the non-normality of data may have implications in the SEM, which needs to be addressed and discussed.

  • The authors present non-normalitty of data as a limitation (lines 734-735), but the real problem here is data analysis and data presentation not being consistent with this, and not the non-normality by itself.

5. The authors report "similar average age for both groups" (line 216) and state that "the groups did not differ in terms of age, BMI or place of study and residence" (lines 306-307). This should be supported with the adequate hypothesis tests.

  • Also, please note that, despite the means are very similar, the two sub-samples present very different standard deviations, and that the standard deviation for the Polish sub-sample is quite weird given the age range.
  • The fact that BMI was computed using self-reported weight and height should be included in the limitations.

6. The development of the Vietnamese versions of the EDI-3 (lines 232-236) and MBSRQ-AS (lines 264-265) was should be better explained, namely regarding some intermediate steps of the process. How was the "best Vietnamese version" selected? Was this selection made for the overall instrument or per item? Besides translation, was any kind of cultural adaptation been made? What were the results of the comparison between the backtranslation with the original questionnaire?

7. In lines 321-323, it is referred that "both groups were dominated by young women slightly above 20 years". This presents two errors, recurrent throughout the text:

  • (1) Terms like "dominated" (besides not being the most adequate) should only be used when referring to proportions, not mean values;
  • In line with this, the idea that "the BMI values were within the normal range" (lines 327-328) should be corrected, and range for BMI in each group presented;
  • The same applies to how the authors report data from other studies (in lines 567-568 does it mean that all participants had normal BMI?);
  • (2) Features which derive directly from the inclusion criteria should not be highlighted as they are in this sentence.
  • Also, please note that the mean value of age presented for the Vietnamese group is not exactly the same in lines 213 (M = 20.25) and 323 (M = 20.24).

8. Considering some of the issues raised above, some parts of the discussion may need to be adapted. The authors should pay special attention to the limitations, considering some of the previous remarks.

9. In the last paragraph (lines 765-774) the authors "point out the pratical implications", but most of what is referred does not derive directly from the current study. This part of the text should present information and implications which highlight the contribute of this particular study to the area, not vague and general considerations.

Author Response

(The authors gave the same response as above.)

Round 2

Reviewer 1 Report

I think the authors have addressed all my concerns and the paper could be considered for publishing. 

Author Response

(The authors gave the same response as above.)

Reviewer 2 Report

The authors have addressed most of the comments and suggestions. Only three issues remain unclear, and should be corrected and/or commented prior to publication.

(1) Regarding sample selection, the main issue was regarding potential selection bias, not the tests used based on the group sizes. Regardless of the biggest group not exceding twice the size of the other, there still may be ocured selection bias, and the fact that only participants from one group were excluded due to "wrong" nationality. These issues should, at least, be considered in the limitations.

(2) In Table 1, despite rank sums provide an adequate description, medians and interquartile ranges are more easily to interpret, and should be preferred.

(3) It was previously noted that "Features which derive directly from the inclusion criteria should not be highlighted", mainly because the authors presented the fact that both groups were "dominated" by women slightly above 20 years, which is due to the inclusion criteria. The reply ("The study is the cross-cultural research. Due to we have decided to discuss all aspects that were analyzed and were proved to have correlations with development of restrictive and bulimic eating behaviors") does not address this adequately

Author Response

(The authors gave the same response as above.)
